# Strengthening COVID-19 pandemic response coordination through public health emergency operations centres (PHEOC) in Africa: Review of a multi-faceted knowledge management and sharing approach, 2020–2021

**Womi-Eteng Oboma Eteng[1]**\*, **Abrham Lilay**[2], **Senait Tekeste**[2], **Wessam Mankoula**[1], **Emily Collard**[3], **Chimwemwe Waya**[1], **Emily Rosenfeld**[4], **Chuck Menchion Wilton**[4], **Martin Muita**[3], **Liz McGinley**[3], **Yan Kawe**[2], **Ali Abdullah**[5], **Ariane Halm**[6], **Jian Li**[7], **Virgil L. Lokossou**[8], **Youssouf Kanoute**[7], **Ibrahima Sonko**[1], **Merawi Aragaw**[1], **Ahmed Ogwell Ouma**[1]

**1** Africa Centres for Disease Control and Prevention, Addis Ababa, Ethiopia, **2** World Health Organisation Regional Office for Africa, Brazzaville, Republic of Congo, **3** Emergency Preparedness, Resilience and Response, UK Health Security Agency, London, United Kingdom, **4** US Centers for Disease Control and Prevention, Atlanta, Georgia, United States of America, **5** World Health Organisation Regional Office for Eastern Mediterranean, Cairo, Egypt, **6** Infectious Disease Epidemiology Department, Robert Koch Institute, Crisis Management Unit, Berlin, Germany, **7** World Health Organisation, Headquarters, Geneva, Switzerland, **8** ECOWAS Regional Center for Surveillance and Disease Control, Abuja, Nigeria

\* etengw@africa-union.org

## Abstract

The coronavirus disease 2019 (COVID-19) pandemic disrupted health security program implementation and incremental gains achieved after the West African Ebola outbreak in 2016 across Africa. Following cancellation of in-person events, a multi-faceted intervention program was established in May 2020 by Africa Centres for Disease Control and Prevention (Africa CDC), the World Health Organisation, and partners to strengthen national COVID-19 response coordination through public health emergency operations centres (PHEOC) utilizing continuous learning, mentorship, and networking. We present the lessons learned and reflection points. A multi-partner program coordination group was established to facilitate interventions' delivery including webinars and virtual community of practice (COP). We retrieved data from Africa CDC's program repository, synthesised major findings and describe these per thematic area. The virtual COP recorded 1,968 members and approximately 300 engagements in its initial three months. Fifty-six webinar sessions were held, providing 97 cumulative learning hours to 12,715 unique participants. Zoom data showed a return rate of 85%; 67% of webinar attendees were from Africa, and about 26 interactions occurred between participants and facilitators per session. Of 4,084 (44%) participants responding to post-session surveys, over 95% rated the topics as being relevant to their work and contributing to improving their understanding of PHEOC operationalisation. In addition, 95% agreed that the simplicity of the training delivery encouraged a greater

**Data Availability Statement:** Data uploaded as supplementary information.

**Funding:** The authors received no specific funding for this work.

**Competing interests:** The authors have declared that no competing interests exist.

number of public health staff to participate and spread lessons from it to their own networks. This just-in-time, progressively adaptive multi-faceted learning and knowledge management approach in Africa, with a consequential global audience at the peak of the COVID-19 pandemic, served its intended audience, had a high number of participants from Africa and received greatly satisfactory feedback.

## Introduction

On March 11, 2020, the World Health Organisation (WHO) declared coronavirus disease 2019 (COVID-19) a pandemic [1]. Implementation of annual program plans of local and international health security stakeholders were abruptly interrupted, testing the recent increases in health security investments that African countries had received since the end of the West African Ebola outbreak in 2016. In-person events including training workshops and on-site support were cancelled to limit the spread of the then-fast-spreading COVID-19 pandemic. As the pandemic progressed, it severely tested existing multi-sectoral coordination structures and mechanisms for health emergency information and resource management by the public health emergency operations centres (PHEOCs) and similar institutional arrangements in countries across the African continent and globally. The pandemic further revealed the need for a whole-of-government approach, necessitating the active participation of an extensive array of stakeholders in the traditional health sector arrangements in the previous, and familiar public health responses.

A PHEOC is a physical location for the coordination of information and resources to support emergency management functions. The functionality of a PHEOC is predicated on four components–policies, plans, and procedures; information system and data standards; skilled human resources; as well as communication technology and physical infrastructure. The timely implementation of functional PHEOCs has been documented as a factor for improved response to emergencies [2,3] and equally important in meeting the minimum requirements of the International Health Regulations (IHR-2005) [4]. As best practice, response resources are arranged and managed using the Incident Management System (IMS), which is a standardised, scalable, and flexible emergency management structure with sets of procedures and protocols that provide a coordinated approach while avoiding duplication of effort for all types of health emergencies.

With the COVID-19 pandemic, programmatic plans and events aimed at strengthening PHEOC capabilities and events including in-person training were disrupted due to the travel and physical distancing measures implemented globally to limit the spread of the pandemic [5]. Virtual systems and communication became the alternative business norm for organising meetings and training delivery. This option became the leverage for delivering webinars accessible through smartphones, tablets, and computers from any part of the world. Several of the available technology platforms had audio and video capabilities [6]. Webinars allowed various interactive opportunities, such as discussion, instant messaging functions, conducting polls, surveys, and knowledge checks. In addition, an interactive and accessible online community of practice could be established alongside webinar training sessions, offering a space to share presentations, continued discussion, peer-to-peer cross-learning and mentoring. Training, learning, and satisfaction through online webinars could lead to excellent outcomes for a significantly large and geographically dispersed audience, compared to face-to-face events, as concluded in a meta-analysis conducted by Gegenfurtner et al. [7].

In the initial months of the COVID-19 pandemic, many Member States in the African continent and globally, where PHEOC capacity was still in development, communicated the need

for operational guidance to partner organisations. In response, a multi-faceted intervention program was established in May 2020 by Africa Centres for Disease Control and Prevention, the World Health Organisation, and partners to strengthen COVID-19 response coordination through public health emergency operations centres (PHEOCs) using continuous learning, mentorship, and networking. The program focused on building and/or strengthening capacity in PHEOC management for a coordinated COVID-19 response and included the creation of an online Community of Practice (CoP) comprising mainly PHEOC professionals from the African continent. This intention was to achieve this through strengthening of attendees and users' capacities and allow them to cascade the acquired competences. The platform was designed to address the immediate knowledge gaps, facilitate the exchange of experience, and serve as a springboard for launching a sustained post-COVID-19 experiential learning base. The primary target audience was personnel working in PHEOCs across the African continent. Besides, participants across the globe were actively engaged throughout the weekly webinars. A complementary Discord (an open-source social communications and online community platform capable of hosting a large virtual group, one-on-one messaging, live in-app audio and video meetings and ease of sharing various file formats) platform Community of Practice was established to provide continuous engagement and networking opportunities [8]. However, no review of the webinar series was conducted to determine whether it had met its objectives, and no lessons learned were drawn to improve future webinar delivery programs.

A review of the webinar series and the Community of Practice (CoP) was conducted to document the experience acquired and to identify recommendations that could potentially serve as a future reference for the establishment of virtual learning programs and CoP within the continent of Africa and in settings similar to Africa.

## Methods

### Program coordination

A core working group was established to lead the overall coordination with members identified from Africa Centres for Disease Control and Prevention (Africa CDC), World Health Organisation Regional Office for Africa (WHO AFRO), World Health Organisation Regional Office for the Eastern Mediterranean (WHO EMRO), WHO Headquarters, UK Health Security Agency (UKHSA), US Centers for Disease Control and Prevention (US CDC), Robert Koch Institute (RKI), European Centres for Disease Prevention and Control (ECDC), Resolve to Save Lives (RTSL) and the global Emergency Operations Centre Network (EOC-NET).

A secretariat was established at the Africa CDC headquarters to oversee program coordination, including scheduling, and coordinating meetings, producing technical briefs, and developing publicity materials (flyers and broadcast emails). The secretariat also provided an enquiry desk to address queries and respond to requests for information and assistance from country teams.

### Webinar facilitation

The webinar series was initiated to respond to identified gaps in multiple aspects of PHEOC operationalisation communicated to WHO and Africa CDC by PHEOC focal points in the initial months of the COVID-19 pandemic. The working group developed a take-off curriculum and content for the webinars delivered through the Zoom communication platform, chosen for its global accessibility, ability to accommodate several hundreds of participants and capabilities such as screen sharing, interactive survey, polls, breakout room and simultaneous translation services. Webinars were generally delivered in English with simultaneous translation into Arabic and French. The webinars were also recorded and broadcast on the Africa

CDC YouTube and Facebook channels to ensure maximum accessibility. Webinars were initially broadcast every week, changing to a biweekly basis in September 2021. Country experience was frequently included in the webinars alongside guest presenters from across the globe. Webinar presenters were typically senior public health specialists, PHEOC professionals, and program leads from across the continent and global public health institutions who were assigned topics based on their areas of expertise.

Special sessions also included live tours of PHEOC facilities, showcasing equipment, human resources, and workflows.

The approach to selecting webinar topics evolved and hence remained dynamic and adaptive. In the initial stages, the first few webinar topics were identified based on the results of an abridged baseline survey that was distributed to those in the PHEOC field across the continent and were known to the PHEOC partners working group. With continuous interactions with webinar participants, through direct feedback, survey findings, and consultations with key actors at country levels, the webinar topics were tailored toward the needs and interests of participants. As the webinars progressed, participants' inquiries into previously treated topics prompted the re-organisation of one-off topics into multi-part series to provide a deep-dive learning experience into various thematic areas of a public health emergency management (PHEM) program—anticipating, preventing, preparing for, detecting, responding to, and recovering from the consequences of public health threats. An emerging best practice for coordinating PHEM programs includes the implementation of functional PHEOCs. Participants who attended at least 75% of the webinar sessions were emailed a link to a knowledge-check in the Flexiquiz platform; those who obtained an 80% pass rate in this knowledge-check were rewarded with an automated certificate of attendance.

## Community of practice

Following an overwhelming interest in the webinar series and in an effort to maintain continuous engagement with experts, networking and response to out-of-topic, but equally important questions, a community of practice (CoP) was formed on Discord on 02 July 2020, one month after the initial webinar. Discord is an open-source social communication and online community platform which was evaluated by the working group to meet the needs of hosting a large virtual community, group, and one-on-one messaging, live in-app audio and video meetings and ease of sharing various file formats [8]. This platform allowed free global access to a PHEOC webinar channel, established by the working group. On this channel, each webinar presentation was uploaded, allowing access to those who could not join the live webinar. Additionally, members of the PHEOC webinar Discord community could share messages, questions, documents, and experiences through the community message board, moderated by members of the webinar working group. Subsequent to this Discord CoP, a PHEOC Network message group was set up on the WhatsApp messaging platform. This message group has been used to share PHEOC-relevant training information, country milestones in PHEM training and situational awareness of emergency responses across the African continent.

## Data variables and analysis

Key indicators were chosen to assess webinar and CoP performance. Relevant registration and participation information were extracted and consolidated for analysis using a pre-defined data extraction tool. For webinars, these included attendance rates from Africa and around the world per webinar, the countries with the most attendees, the length of the webinar sessions, and the number of participants-to-facilitator discussions.

Participants' responses to the post-webinar polls, which began during webinar 16 and were administered until webinar 52, were reviewed, with an additional question to clarify the attendee's role in the PHEOC from webinar 30 to 52. The questions were:

- Will you attend another webinar based on the delivery of this session?

- Did the session improve your understanding of the topic?

- Was the content easy to follow?

- Was there enough time for questions and discussion?

- Would you recommend the session to someone?

- How relevant was the topic to your current role in the COVID-19 response?

- Is your current role related to Public Health Emergency Operations Centre?

A thematic analysis (inductive, and semantic approaches) was used to group, analyse, and describe major findings. Further descriptive statistics (percentages, tables, charts, etc.) were used to summarize the quantitative data. MS Excel and R [9] were used to analyse the quantitative data and produce tables and figures.

### Ethics statement

The data collected in the frame of this study did not involve any personally identifiable information and data was anonymised. Respondents invited to fill in the online survey feedback were informed about its purpose and objectives and were aware by providing their answers they consented to participate.

## Results

### Establishing the webinar series

**Baseline survey.**   In May 2020 before the first webinar, an initial baseline survey was created using an online survey tool and shared with stakeholders from across the PHEOC networks. The responses to the survey were used to help develop and refine the topics and content of the webinars.

The summary of survey responses showed the working group what challenges Member States were facing and areas of good practice in PHEOCs responding to COVID-19 (Table 1).

**Webinar planning and coordination.**   The webinar coordination group had the responsibility for developing and agreeing on the webinar topics, schedule, and content. Decisions were generally made by consensus, and where not possible such decisions were arrived at by a simple majority. Through direct feedback, survey findings, and consultations with key actors at the country level, the webinar topics were tailored towards the needs and interests of participants. The secretariat was then responsible for the official marketing of the webinars with the development of a flyer which was shared through agreed PHEOC networks, email distribution lists, and word of mouth.

**Quality assurance of webinar content.**   The webinar coordination group was responsible for ensuring consistency of presentations with established terminologies, workflow, and alignment with existing guidance documents, except where innovative approaches were considered. Presenters were required to share final presentations along with any related reference documents prior to the webinar for review and agreement by the working group.

**Table 1. Summary response of baseline survey, May 2020.**

| Summary of the Challenges | Areas of Good Practice |
| --- | --- |
| • Resilience and maintaining the response long term–supplies, funding, and human resources.<br>• Managing concurrent incidents and emergencies in addition to COVID-19<br>• Staff welfare (fatigue) and reducing the exposure risk.<br>• Cross Government involvement and unfamiliarity with public health response structures and PHEOC principles<br>• Lack of funding, human resources, and supplies<br>• Leadership untrained or unfamiliar with PHEOC principles<br>• Command, control, and coordination<br>• Cross-regional and other stakeholder collaboration<br>• Basic communications systems and other infrastructure | • Activation of PHEOC improved coordination and unity<br>• Multi-level response allows for regional particularities and specifics to be considered and addressed.<br>• Use of event bases surveillance and early warning systems benefited preparedness and global tracking of COVID.<br>• Concurrent incident management<br>• Multi-sector and partner involvement<br>• COVID-19-specific government-level input<br>• Served as a platform for cooperation cross-border and cross-boundary where previous political divisions exist |

## Webinar topics and series

During the early phases, topics for the webinars were identified and prioritised by the core team using a majority vote system to identify and prioritise topics. However, survey findings and consultations with key public health actors at the country level including national public health institutions and partners supporting emergency management were used to inform webinar topics tailored to the interests of participants in later months. Some of the prioritises topics were reorganised into series where they were covered in subsequent several sessions under each series to allow participants to take a deep dive into the content and gain a thorough understanding of the respective topic.

Based on the findings from this review, the average webinar length was 1.8 hours, with sessions lasting anywhere between 1.5 and 2.7 hours. During the webinar period, 56 topics were covered across eight thematic areas. Each of the IMS and incident leadership themes had 14 topics. The information management series covered seven topics, while the multi-sectoral coordination, PHEOC handbook, and legal framework series each covered five topics. Other series had four or fewer topics (Fig 1).

Development of PHEOC document part 2, IMS round-up, and development of PHEOC documents part 1 sessions were attended by 551 (4.3%), 466 (3.7%), and 382 (3.0%) participants, respectively. On the other side, 47 (0.4%), 25 (0.2%), and 12 (0.1%) attended Watch Mode operations: EBS tools, PHEOC information systems part 3, and PHEOC information systems part 4 webinar sessions, respectively (Fig 2 and S1 Table).

## Webinar participation

A total of 56 webinar sessions were held between June 2020 and December 2021, with initial sessions held on Thursdays, starting at 3:00 p.m. East Africa Time (EAT). However, an adjustment was made later to accommodate more participants and pushed the start time to 4 p.m. EAT.

S2 Table shows that a total of 95,230 participants were registered, with 12,715 (13%) unique attendees from 130 countries across the globe participated in at least one live webinar session. An 85% return rate (number of participants who attended more than one session) was recorded (uniquely identified by username and email address). Of those who attended, 8,528

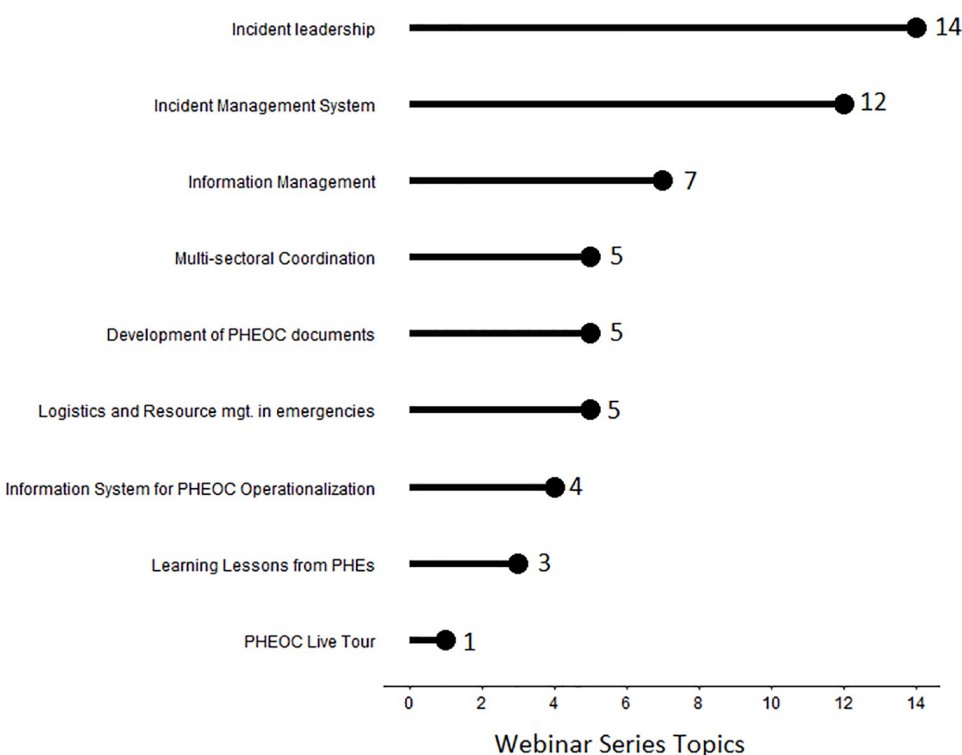

**Fig 1. Webinar Series thematic areas and topics, 2020–2021.**

(67%) were from the African continent. All 55 countries in the continent participated in at least one webinar. A total of 48 countries were represented in at least half of the 56 webinars with 36 (65%) coming from the African continent. Over the webinar period, an average of 33 with a range of 8 to 50 countries participated from the African continent (Fig 3). A cumulative 1,102,354 views of the webinars were recorded across Facebook and YouTube platforms where the live webinars were simultaneously broadcasted.

Seventeen countries were identified throughout the webinar periods as having the top three attendance rankings based on the number of attendees each week, with 13 being from the African continent. Nigeria, the United States of America (USA), Kenya, and Ethiopia accounted for 45% of all webinar attendees (Fig 4). Across all live webinar sessions, there was almost twice (1.7) the number of attempts to join the sessions compared to actual participation.

## Post-webinar survey

During the period when the post-webinar surveys were administered starting with webinar 16, 4,084 (44%) of the webinar participants (9,283) responded. Over 95% responded positively to the topic's relevance to their current role, the likelihood of recommending a topic to another colleague, the session's contribution to improving their understanding of the issue, interest in attending another session, and that the content was easy to understand. Furthermore, 85% said there was enough time for discussion at the end of the sessions. An additional question was added on their role related to PHEOC and 2,404 (81%) of the 2970 webinar participants responded that they had a PHEOC-related role (Table 2).

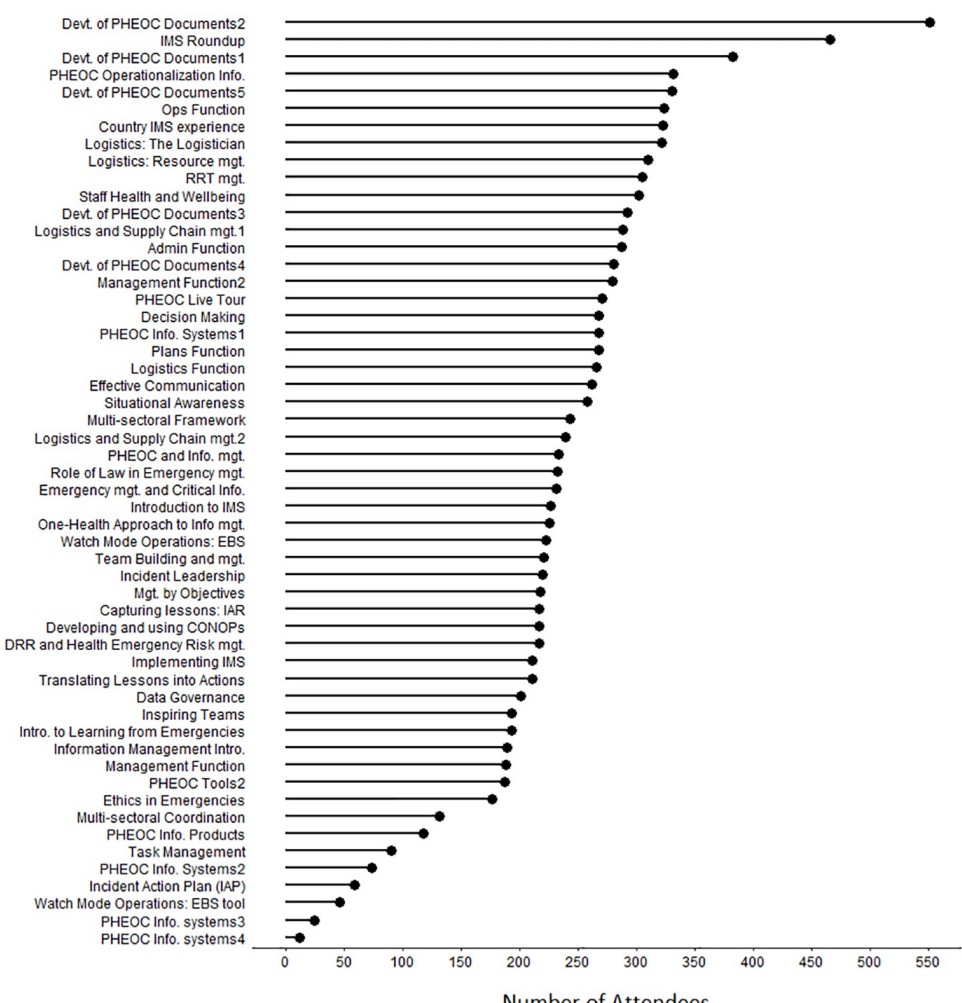

**Fig 2. Webinar individual topics and Attendees, 2020–2021.**

## Interaction and networking within the online CoP

In all the webinar sessions, participants had an opportunity to interact with other participants and facilitators or to ask a question and obtain an answer (Q&A) and share general opinions regarding the topic of the presentation and other areas related to PHEOC operations. There were 1,407 total interactions between webinar participants and facilitators. On average, there were 26 interactions between the participants and facilitators, ranging from 4 to 56 interactions per session.

The 'Discord Platform' was acknowledged by members of the online CoP to be an effective platform that enabled continuous engagement and networking across Member States and increased their knowledge and skills on the topics addressed. It facilitated peer-to-peer learning through country-to-country, PHEOC-to-PHEOC communication about implementing coordinated COVID-19 responses. The platform was utilised by participants, before, during, and after webinars sessions, to discuss real-time issues and needs (e.g., advocacy for decision-makers commitment, PHEOC funding, and legal framework development, etc.), sharing of experience, lessons learned and best practices (e.g., PHEOC-to-PHEOC mentorship on

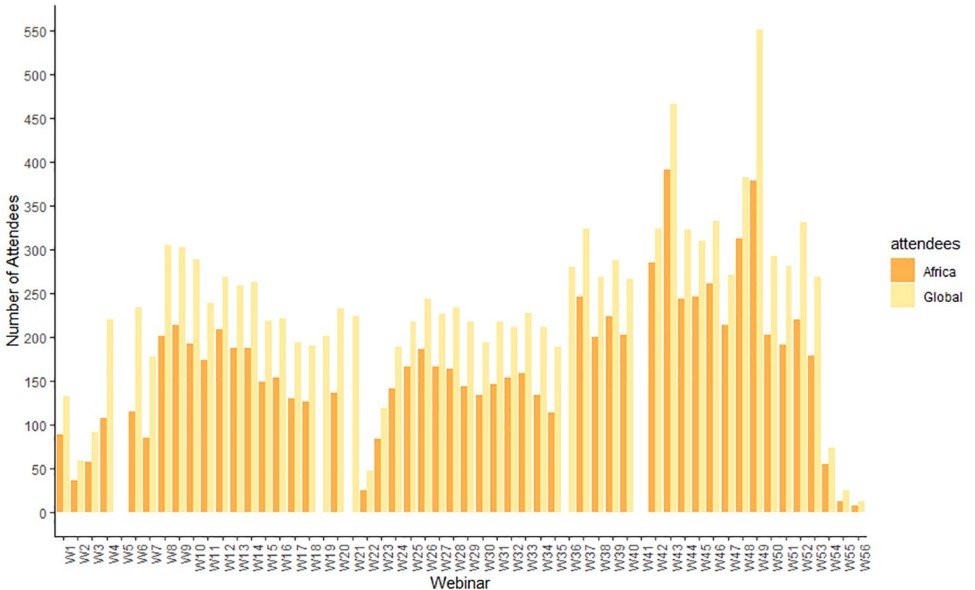

**Fig 3. Webinar Global and African continent Attendees, 2020–2021.**

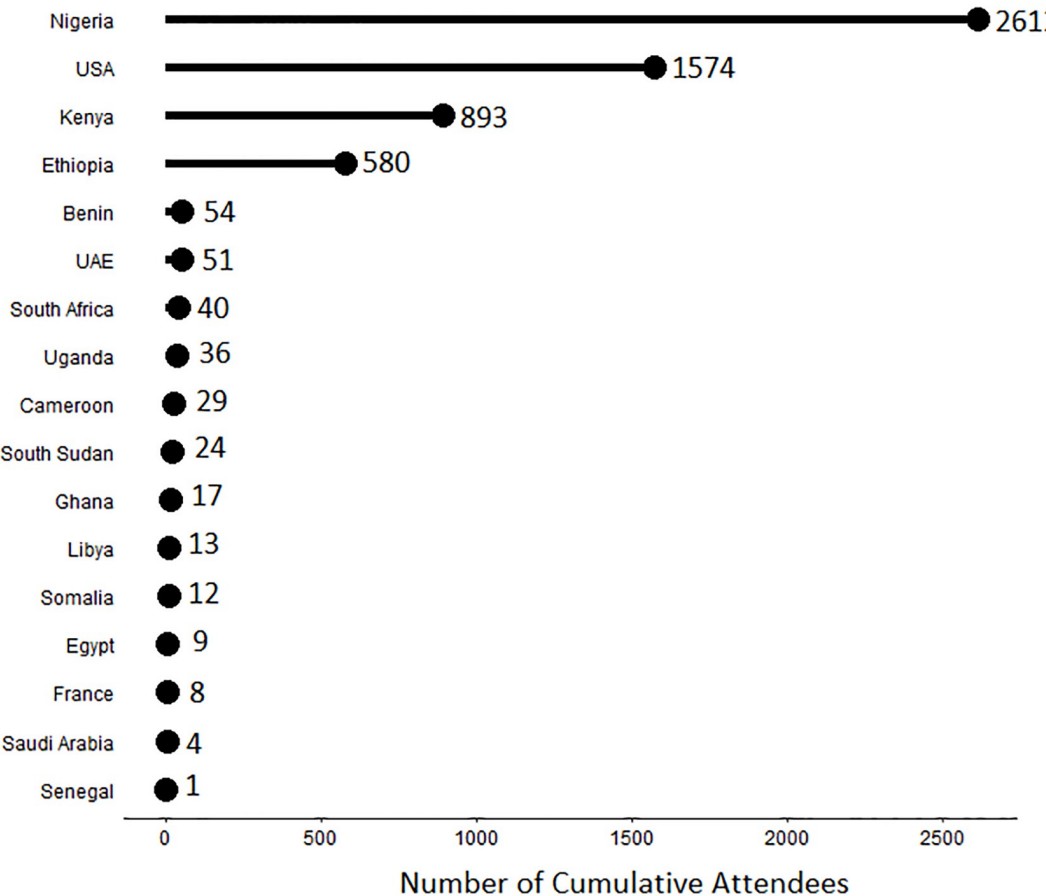

**Fig 4. Webinar attendees top represented countries, 2020–2021.**

**Table 2. Responses of post-webinar surveys, 2020–2021.**

| Question | Yes | % | No | % | Total (N = 4,084) |
|---|---|---|---|---|---|
| Topic relevant to the current role | 4,034 | 98.8 | 50 | 1.2 | 4,084 |
| Would recommend the session to someone else | 4,024 | 98.5 | 60 | 1.5 | 4,084 |
| The session improved understanding of the topic | 4,009 | 98.2 | 75 | 1.8 | 4,084 |
| Would attend another session | 3,970 | 97.2 | 114 | 2.8 | 4,084 |
| The content was easy to follow along with | 3,901 | 95.5 | 183 | 4.5 | 4,084 |
| Adequate time for questions and discussion | 3,470 | 85.0 | 614 | 15.0 | 4,084 |
| The current role relates to PHEOC | 2,404 | 80.9 | 566 | 19.1 | 2,970 |

potential PHEOCs support in COVID-19 response), and required documentations for PHEOC operationalisation (e.g., sharing of relevant PHEOC manuals, SOPs, training materials and events, and other useful references).

As of September 2022, approx. 1789 online community members were found to be active on the platform, which is still active. The webinar coordination group members continuously work to improve and ensure that the online CoP and the virtual training through the Webinar Series endure beyond the COVID-19 pandemic.

## Reward on participation

To encourage participation and intake of training as well as to reward active participants, e-certificates were provided to those who attended at least four webinar sessions and successfully completed the knowledge-check with an 80% pass rate. A total of 2,102 participants, 15% of those who attended the certificated series, received e-certificates, with logos of all coordinating partners.

## Discussion

The study presents findings from the review of a multifaceted approach for virtual networking and learning focused on PHEOC capacity building from 2020–2021. The establishment of such a webinar series and CoP platform was aimed at providing just-in-time training in PHEOC operationalisation as hubs supporting the implementation of coordinated COVID-19 responses across Africa.

The outcomes of the review showed the virtual learning and networking platform served the intended primary audience with an extensive global reach, drawing in participation from over 130 countries, including all 55 African countries. The Webinar series significantly contributed to building and/or strengthening the knowledge of PHEOCs professionals through active networking, and country-to-country sharing of experience, lessons learned and best practices, and required documentation for implementing PHEOCs and coordinated COVID-19 responses. However, these findings should be interpreted with caution due to the limitations of the review and Webinar series.

There were sufficient registrants (over 95,000) interested in joining the PHEOC Webinar Series. However, there was an overall low attendance rate (13%), and close to half (47%) of the total webinar participants came from just 17 countries. This high registration rate may reflect the genuine need and interest in PHEOC management material at the time. Conversely, the low attendance rate may reflect several scenarios including a lack of time to attend a 1–2-hour long webinar during the COVID-19 pandemic, particularly for those working in a PHEOC, a lack of technological infrastructure to join the webinar online and conflicting commitments at

the time of the webinars. However, the relevance of the webinars and the interest it garnered over time was demonstrated by a high return rate of 85% (that is, the percentage of unique attendees who participated in more than one live session). There was nearly twice (1.7) the number of attempts to join live sessions compared to actual participation. Recordings of the sessions were distributed by email and on the Discord platform to carter for this challenge, likely due to low internet bandwidth connectivity. Among the motivation strategies employed, the incorporation of attendance certificates to promote participation and engagement, based on the successful completion of a knowledge check, did not appear to act as a significant factor in webinar participation, with just 15% of those who attended at least 75% webinar sessions successfully obtaining a certificate.

Most (67% of 12,715) of the attendees across all webinars were from the African continent (Benin, Cameroon, Egypt, Ethiopia, Ghana, Kenya, Libya, Nigeria, Senegal, Somalia, South Africa, South Sudan, and Uganda). Nigeria, the United States of America (USA), Kenya, and Ethiopia accounted for 45% of all webinar attendees. This strong representation from the African continent and the United States could be interpreted on several levels including that there was a greater need for information and learning on PHEOC management and operationalisation from these countries. However, the inclusion of key partners of Africa CDC and US CDC in the PHEOC Webinar Series working group is likely to have resulted in an inordinately large number of participants from outside the Africa PHEOC network.

The adaptive and evolving nature of the webinar program helped to improve the delivery of the sessions and tailored it to the interests of participants across the continent and beyond. The average number of attendees (235) per session and approximately 70% of the webinar sessions had at least 200 attendees indicating a successful uptake of the webinars. This could be because the webinar topics were tailored to their interests and the fact that there were increasingly strong planning and coordination efforts between the working group and attendees/ community of practice as the webinars continued.

Inviting countries to share their experiences in the management of COVID-19 helped other national PHEOC staff learn from peers and adapt the lessons in their context and to engage in an active exchange with other experts in the region. Webinar participants particularly engaged with sessions which focused on sharing experiences from Member States on their coordination through PHEOC. Through the process, 17 countries from Africa and several from outside the continent had a chance to share their experiences. The PHEOC webinar sessions containing the country experience sharing were mainly provided with the help of PowerPoint presentations by subject matter experts (SMEs) from invited countries and various public health organisations. Previous studies have indicated that most resource persons (74%) successfully delivered lectures during online capacity building with help of PowerPoint presentations [10].

Based on the findings from this review, the average webinar length was 1.8 hours, with sessions lasting anywhere from 1.5 to 2.7 hours, this reflected the complexity of the topic discussed and the engagement of the attendees in the subsequent Q&A session. Organisers recognised that participants tended to drop off the webinar after one hour, likely because of conflicting commitments. Therefore, it is recommended that future webinar sessions do not extend beyond one hour. Where it is likely to last longer up to 1.5 hours, it may be beneficial to inform participants ahead of time and keep them actively engaged [11]. In terms of the time taken per session, 85% of attendees agreed that enough time was allotted at the end of the webinar session for discussion and answering questions. There were 1,407 interactions between webinar participants and organises during the webinar period, including Q&A and opinion sharing, resulting in an average of 26 interactions per session. The interactions observed appear to be low as compared to the number of attendees per session (235 on average), which could be due to the limited time allocated for discussion. However other considerations may

include the likelihood of attendees listening to the session while conducting other work, language barriers or their ability or confidence to engage with the other attendees and facilitators in a virtual environment.

The PHEOC webinar sessions were held on Thursdays, taking the recommendations of a study that attributed better participation in attending webinars on Tuesdays, Wednesdays, and Thursdays, these being the best days for live events in countries where Saturday and Sunday are the weekends [12]. In the first 12 months of the series, sessions were conducted at the same time each Thursday, on reflection and based on feedback from participants, it was decided that weekly sessions should use an alternating time to mitigate the chance of a clash with re-occurring pre-existing meetings.

Almost all respondents (over 95%) to the post-webinar surveys agreed that the topics of the webinar sessions were relevant, the contents were easy to comprehend and contributed to improving their understanding. Similar studies (91%) agreed that webinar topics were relevant [10]. The webinar coordination group met regularly (virtually) and were in frequent contact via email to discuss and refine the weekly content, using expertise from across all partner organisations. Likewise, the group was open to suggestions from international colleagues and attendees regarding on-topic content and presentation of content. An active and efficient secretariat was crucial in the development and distribution of Webinar brochures and links which were communicated via registrants' email and published on the websites of the Africa CDC. Previous studies have indicated improved engagement (89%) of respondents when organisers provide webinar links well in advance for registration [10]. In addition, another study found that webinar participants (75%) were interested in the topic rather than the speaker or the company organising the webinar [12].

The success of the topic choice and strong communication of webinar sessions in advance is indicative of the commitment of the PHEOC Network partners, PHEOC Webinar working group, and secretariat. They emphasised the effort involved in conducting a weekly, interactive webinar series, especially when hosted by SMEs who themselves were engaged in COVID-19 response activities. The amount of effort involved in organizing similar public health learning tools/models should be taken into account by planning teams, and consideration given to the resources available, the ambitiousness of the scale of the learning program, and the level of commitment required.

## Limitation

The low response rate of webinar surveys may have influenced the topic selection process, as those who did not vote may have preferred a different topic. Furthermore, missing data on participants' relevant work experiences may have an impact on translating the learning into practice, as those who voted may not necessarily hold critical roles in the PHEOC.

## Conclusions

In conclusion, the 'just-in-time' multifaceted approach designed to help address the gap in PHEOC capabilities during the initial days of the COVID-19 pandemic, provided many lessons for the organisers and reflection points for those embarking on similar initiatives. Key lessons included the scope of planning involved in such an approach; the logistical efforts required to ensure that presenters, facilitators, translation services and participants were present at each webinar; the engagement involved to ensure that the webinars and CoP platform were effective; the inclusion of a wide target audience; the potential of increasing participation of those registered to attend the webinar series; the evaluation of the approach and how it met the needs of PHEOC information sharing and strengthening. While the organisers recognise

the limitations of such an approach, feedback from those who participated in the Webinars and CoP have been largely positive, with many of those participants continuing to engage in the PHEOC CoP, particularly from the African continent. The virtual approach used in delivery, bolstered by the changing attitudes to learning and working across online global platforms, proved useful for PHEOC knowledge sharing and we recommend that further investment should be considered in virtual learning platforms for PHEOC training in the Africa continent and in similar settings. This would allow swift implementation and up-scaling of alternative capacity strengthening and exchange approaches for public health staff leading response operations in Africa, when presential and on-site technical support and mentoring might not be possible to the extent required.

## Supporting information

**S1 Table. Webinar thematic areas, and topics by webinar registrants and attendees, 2020–2021.**
(PDF)

**S2 Table. Frequency of attendance, Webinar series, 2020–2021.**
(PDF)

**S1 Data. Database Analysis PHEOC Webinars 2022.9.26.**
(XLSX)

## Acknowledgments

We would like to acknowledge all countries that attended the webinar sessions and participants for completing the feedback survey and for continued support throughout the webinar period.

## Author Contributions

**Conceptualization:** Womi-Eteng Oboma Eteng, Abrham Lilay.

**Data curation:** Womi-Eteng Oboma Eteng, Abrham Lilay, Senait Tekeste, Wessam Mankoula, Emily Collard, Chimwemwe Waya, Emily Rosenfeld, Chuck Menchion Wilton, Martin Muita, Liz McGinley, Yan Kawe, Ali Abdullah, Ariane Halm, Jian Li, Virgil L. Lokossou, Youssouf Kanoute, Ibrahima Sonko, Merawi Aragaw, Ahmed Ogwell Ouma.

**Formal analysis:** Womi-Eteng Oboma Eteng, Abrham Lilay.

**Investigation:** Womi-Eteng Oboma Eteng, Abrham Lilay, Senait Tekeste, Wessam Mankoula, Emily Collard, Chimwemwe Waya, Emily Rosenfeld, Chuck Menchion Wilton, Martin Muita, Liz McGinley, Yan Kawe, Ali Abdullah, Ariane Halm, Jian Li, Virgil L. Lokossou, Youssouf Kanoute, Ibrahima Sonko, Merawi Aragaw, Ahmed Ogwell Ouma.

**Methodology:** Womi-Eteng Oboma Eteng, Abrham Lilay, Senait Tekeste, Wessam Mankoula, Emily Collard, Chimwemwe Waya, Emily Rosenfeld, Chuck Menchion Wilton, Martin Muita, Liz McGinley, Yan Kawe, Ali Abdullah, Ariane Halm, Jian Li, Virgil L. Lokossou, Youssouf Kanoute, Ibrahima Sonko, Merawi Aragaw, Ahmed Ogwell Ouma.

**Supervision:** Wessam Mankoula.

**Validation:** Womi-Eteng Oboma Eteng, Abrham Lilay.

**Visualization:** Womi-Eteng Oboma Eteng, Abrham Lilay, Liz McGinley.

**Writing – original draft:** Womi-Eteng Oboma Eteng, Abrham Lilay.

**Writing – review & editing:** Womi-Eteng Oboma Eteng, Abrham Lilay, Senait Tekeste, Wessam Mankoula, Emily Collard, Chimwemwe Waya, Emily Rosenfeld, Chuck Menchion Wilton, Martin Muita, Liz McGinley, Yan Kawe, Ali Abdullah, Ariane Halm, Jian Li, Virgil L. Lokossou, Youssouf Kanoute, Ibrahima Sonko, Merawi Aragaw, Ahmed Ogwell Ouma.

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
