## [Decision Letter · Decision Letter 0]

2 Jan 2023

PGPH-D-22-01832

Strengthening response coordination through public health emergency operations centers in Africa: Lessons learned from 56-week webinar sessions, 2020-2021

Dear Womi Eteng

Thank you for submitting your manuscript to PLOS Global Public Health. After careful consideration, we feel that it has merit but does not fully meet PLOS Global Public Health’s publication criteria as it currently stands. Therefore, we invite you to submit a revised version of the manuscript that addresses the points raised during the review process.

Please submit your revised manuscript by February 15. If you will need more time than this to complete your revisions, please reply to this message or contact the journal office at globalpubhealth@plos.org. Please include the following items when submitting your revised manuscript:

We look forward to receiving your revised manuscript.

Kind regards,

Megan Coffee, MD, PhD

Academic Editor

Journal Requirements:

1. Please amend your online Financial Disclosure statement. If you did not receive any funding for this study, please simply state: “The authors received no specific funding for this work.”

2. Please update your online Competing Interests statement. If you have no competing interests to declare, please state: “The authors have declared that no competing interests exist.”

3. In the online submission form, you indicated that "Data could be accessed upon request to the author". All PLOS journals now require all data underlying the findings described in their manuscript to be freely available to other researchers, either 1. In a public repository, 2. Within the manuscript itself, or 3. Uploaded as supplementary information.

4. We do not publish any copyright or trademark symbols that usually accompany proprietary names, eg (R), (C), or TM  (e.g. next to drug or reagent names). Please remove all instances of trademark/copyright symbols throughout the text, including © on page 7.

5. Please include a separate legend for each figure in your manuscript.

Additional Editor Comments (if provided):

Reviewers' comments:

Reviewer's Responses to Questions

**Comments to the Author**

1. Does this manuscript meet PLOS Global Public Health’s publication criteria? Is the manuscript technically sound, and do the data support the conclusions? The manuscript must describe methodologically and ethically rigorous research with conclusions that are appropriately drawn based on the data presented.

Reviewer #1: Yes

Reviewer #2: Partly

2. Has the statistical analysis been performed appropriately and rigorously?

Reviewer #1: Yes

Reviewer #2: Yes

3. Have the authors made all data underlying the findings in their manuscript fully available (please refer to the Data Availability Statement at the start of the manuscript PDF file)?

Reviewer #1: Yes

Reviewer #2: Yes

4. Is the manuscript presented in an intelligible fashion and written in standard English?

Reviewer #1: Yes

Reviewer #2: Yes

5. Review Comments to the Author

Reviewer #1: This is an interesting paper, informative and well written. However I feel a little more analysis and some slight edits are needed

Line 122: data extraction too (correct to tool)

Line 194 - 197: A table // annex describing what these thematic / topics are and contain would be helpful. It isn't clear what is contained in PHEOC doc part 2 for example

Line 212 - Please describe what a 'discord platform' is

Line 251 - Nigeria and Ethiopia participation - authors should consider whether the populations of the countries (the two biggest in Africa) is a part explanation for the large participation. Further reflection on why such large US participation since US was not a target

Line 269-279 - the sentences here seem incomplete

General

Some information on languages used for delivering the webinars and reflection on the impact on participation would improve this review. Africa Union has 5 official languages, what proportion of sessions were delivered in English, French Arabic, Portuguese etc? A correlation between language of delivery and country participation may be informative

Impact on practice - some comment on the impact of the webinar series on actual practice - operationalization of PHEOCs would be informative. If this has not been done, perhaps the authors could indicate why not and how they propose to do this in the future to inform further development of the programme?

Reviewer #2: Thank you for the opportunity to review this paper, which examines the implementation of a knowledge support service by an inter-agency team during the COVID-19 pandemic. The intervention aimed to strengthen COVID-19 response of Public Health Emergency Operations Centres.

The adaptation of knowledge support and information dissemination practices are critical changes that took place during the pandemic, and which have had wide-reaching implications for public health interventions and crisis response. The intervention that is presented in this paper can provide an important addition to the literature, to improve our understanding of how to effectively adapt in a changing context with movement restrictions. The evaluation presented can help to inform agencies and organisations seeking to do similar work.

At present, the organization of the paper and also the content needs some revision to be able to provide clear evidence. In its current format, the manuscript provides a mix of information in the introduction, methods, results, and discussion, with some results described in the introduction, the methods are very short, some methods are described in the results, and some results are presented for the first time in the discussion. Also, the title implies that the paper reviews the webinar series, but the paper provides some interesting information about a Community of Practice, which is useful and important to share. The title should be revised to reflect this or perhaps to describe a multi-faceted intervention to share knowledge and experiences for teams engaged in the COVID-19 response (e.g. webinar series and CoP).

Another key issue is that the findings presented do not examine quality and it isn’t clear if it was possible to examine effectiveness or impact of the webinars (e.g. accurate knowledge acquisition and change in practice). In order to receive a certificate, participants had to attend all webinars and also complete a knowledge check. If the results of the knowledge check are available, this would help to provide some insight into quality.

The examination of the results and limitations is currently a bit superficial with descriptive statics, and the conclusions about effectiveness aren’t supported by the presented findings. I’ve provided some detailed comments below which I hope will be helpful.

Overall organization – The introduction should be used to provide information about the context and the need/justification for developing the intervention. The methods should describe what the author group has done (e.g. how was the intervention assessed/evaluated?). The results section should describe the intervention itself and any analyses that were possible. All components of the intervention – including the development of the Community of Practice (CoP), with clear descriptions of the aims and objectives for each component, should be included. The discussion should unpack the meaning of the results and consider findings of other relevant studies or interventions. It might also be helpful to provide a timeline which indicates the declaration of the pandemic and key interventions that this study examines (forming of secretariat, intervention planning, webinar start date, set up of CoP, start date of including participants’ inputs into the topics, start date of introducing language translation (if done), webinar finish date). It is also necessary to clarify who the different actors and contributors are. Is the PHEOC working group specific to this webinar series? Or the CoP? Or is it broader?

Page 4 line 122 – typo. “Data extraction tool” is missing an l at the end of “tool”

Page 4 Results – please indicate when the intervention was held (date range of webinars, date or month when the CoP was set up, etc)

Page 5-6 Table 1 – for the discussion, it would be interesting to unpack the overlaps in the “Summary of Challenges” and “Areas of Good Practice” as several topics appear in both.

Page 6 lines 160-162 – this information is presented earlier. Suggest to remove the earlier information and keep it here in the results section

Page 6 lines 170-174 – quality assurance. This is a crucial topic that should be significantly expanded. It is particularly important in the rapidly evolving context of the pandemic. Did the organizers succeed in obtaining and reviewing all presentations before they were given? Was the terminology kept consistent? Or were corrections made during the course of individual webinars? Were there any challenges in the quality assurance activities? Any gaps? Lessons learned?

Page 6 lines 175-183, Webinar delivery – was language translation done? It sounds like it may have been but it isn’t clear for which languages and the text implies it wasn’t done for all the sessions. This is a huge issue for reach and comprehension – and therefore quality, safety and effectiveness of the intervention – and should be addressed. Also were the surveys sent in multiple languages? Does the CoP have some support for translation or what is/are the functional languages?

Page 7 line 187 – who are the public health actors at country level? Please clarify which stakeholders were include in the design and content of the webinars

Page 7 line 189 – were the deep dive format of webinar for 1.5-2 hours enough to provide a “thorough” understanding? This statement will need to be defended or the language should be altered.

Page 7 paragraph 3 – the contents of this paragraph would be more effectively presented in a table. The table should list all the themes (column 1) and all the subtopics for each theme (column 2). It would help to also provide an additional column with number and % participants for each subtopic to get an idea of uptake and interest. Also consider including a column for the number of people who registered, to look for patterns in registration vs attendance. The text should examine the table and the patterns in uptake to help characterize the update and participation.

Page 7 – post-webinar survey- why was the period of post-webinar surveys limited? Why wasn’t it done for all of them? Was it delivered in multiple languages?

Page 8 paragraph 1 (Interaction and networking within the online CoP) – this paragraph is difficult to follow and should be reworked to clarify and improve the flow of the text. As mentioned above – this is a separate component of the intervention – with a different aim, different objectives and different delivery/management than online webinars and perhaps also a different time period (is the CoP still active?). Also, please explain with the “Discord Platform” is, as this is not widely known.

Page 8 Reward on Participation – the knowledge check is briefly mentioned here. If available, the findings/scores of the knowledge check should be presented to provide some idea of participant knowledge. If any pre-tests were done, then the pre- and post-tests should be compared.

Page 8 Discussion lines 231-2 – this is the first place where the reader learns that the webinars were biweekly. This should be put into the results in a description of the intervention.

Page 8 line 236 “outreached” – this is an unusual use of this term, would be more effective to rephrase. Suggest “reached beyond the intended primary audience” or something to that effect.

Page 8 line 237 – this conclusion is not supported by the data. Participants’ perceptions suggest the webinar strengthened knowledge but there isn’t any objective evidence to prove this happened. There is also no clear examination of the CoP impact on knowledge, access to information or networking.

Page 9 paragraphs 2-5. The discussion should critically examine the patterns in the findings, such as the changing pattern in attendance. There appeared to be higher uptake, then a lull and then a steady increase later on (Figure 1). Why did this happen? What changes were made in implementation that might have affected this? How could communication or advertising have impacted it? What about language, any concurrent discussions in the CoP, or the ability of participants to influence the topic? These and other factors may have affected participation in different seminars; it would be helpful to understand whether or not that might be the case and any evidence to support.

Page 9 lines 264-265 – Is there evidence that inviting countries to share experiences in this intervention helped other countries learn, adapt and engage? If so, it should be presented here. All conclusions must be supported by the findings.

Page 10 paragraph 2 – the second half of the paragraph presents the findings and should be put into the results section

Page 10 paragraph 3 – were these surveys done in English? How would that affect the reach and validity of the findings?

Page 10 lines 290-291 – the sentence is unclear, please rework to clarify

Page 10 lines 293-295 – This implies success when there appears to be variable uptake of the webinars. This is a good space to critically examine the findings.

Page 11 – Limitations – this section should be markedly expanded. My previous comments allude to a number of limitations that should be described. An especially important limitation is the lack of measurements for quality and impact of the webinars as well as the CoP. It’s a huge challenge for all knowledge and capacity strengthening programmes, whether online or in-person, and should be unpacked here and any activities/adaptations that were made to address the limitations should be described. Please also note that the survey is likely to be prone to response bias and pleasing.

Page 11 – Conclusions, line 315-318 – the findings presented do not support this conclusion. If there are data to support the conclusion, this should be included in the results and unpacked in the discussion.

6. PLOS authors have the option to publish the peer review history of their article (what does this mean?). If published, this will include your full peer review and any attached files.

**Do you want your identity to be public for this peer review?** For information about this choice, including consent withdrawal, please see our Privacy Policy.

Reviewer #1: **Yes: **Dr Ebere Okereke

Reviewer #2: No

---

## [Decision Letter · Decision Letter 1]

23 Mar 2023

PGPH-D-22-01832R1

Strengthening COVID-19 pandemic response coordination through public health emergency operations centers in Africa: Review of a multi-faceted knowledge management and sharing approach, 2020-2021

Dear Mr Eteng

Thank you for submitting your manuscript to PLOS Global Public Health. After careful consideration, we feel that it has merit but does not fully meet PLOS Global Public Health’s publication criteria as it currently stands. Therefore, we invite you to submit a revised version of the manuscript that addresses the points raised during the review process.

I would also streamline the discussion section and include the advice the reviewers have outlined in detail below.

Please submit your revised manuscript by May 1st. If you will need more time than this to complete your revisions, please reply to this message or contact the journal office at globalpubhealth@plos.org. Please include the following items when submitting your revised manuscript:

We look forward to receiving your revised manuscript.

Kind regards,

Megan Coffee, MD, PhD

Academic Editor

Journal Requirements:

2. In the online submission form, you indicated that "Data could be accessed upon request to the author". All PLOS journals now require all data underlying the findings described in their manuscript to be freely available to other researchers, either 1. In a public repository, 2. Within the manuscript itself, or 3. Uploaded as supplementary information.

Additional Editor Comments (if provided):

Reviewers' comments:

Reviewer's Responses to Questions

**Comments to the Author**

1. If the authors have adequately addressed your comments raised in a previous round of review and you feel that this manuscript is now acceptable for publication, you may indicate that here to bypass the “Comments to the Author” section, enter your conflict of interest statement in the “Confidential to Editor” section, and submit your "Accept" recommendation.

Reviewer #3: (No Response)

Reviewer #4: (No Response)

Reviewer #5: All comments have been addressed

2. Does this manuscript meet PLOS Global Public Health’s publication criteria? Is the manuscript technically sound, and do the data support the conclusions? The manuscript must describe methodologically and ethically rigorous research with conclusions that are appropriately drawn based on the data presented.

Reviewer #3: Yes

Reviewer #4: Yes

Reviewer #5: Yes

3. Has the statistical analysis been performed appropriately and rigorously?

Reviewer #3: I don't know

Reviewer #4: Yes

Reviewer #5: Yes

4. Have the authors made all data underlying the findings in their manuscript fully available (please refer to the Data Availability Statement at the start of the manuscript PDF file)?

Reviewer #3: Yes

Reviewer #4: Yes

Reviewer #5: Yes

5. Is the manuscript presented in an intelligible fashion and written in standard English?

Reviewer #3: Yes

Reviewer #4: No

Reviewer #5: Yes

6. Review Comments to the Author

Reviewer #3: It is important to have this abbreviation PHEOC in the title as it appears in the short title => Strengthening COVID-19 pandemic response coordination through public health emergency operations centers (PHEOC) in Africa: Review of a multi faceted knowledge management and sharing approach, 2020-2021

Use the same language (British English or American English). For instance, the word “Centers” in the title but written as “Centre” in the key words section.

Abstract:

There must be an omission in the following sentence: In-person events including training workshops and on-site technical support were canceled to limit the spread of the ??? then fast-spreading COVID-19 pandemic. Or good to delete the word “then”. Method: Put full stop after the first sentence. The abbreviation PHEM is not defined at its first appearance.

Introduction:

There is a need to define what is a functional PHEOC and what are the criteria set to judge it functional. The text from line 86 to 112 seems describing the methods used and could be in the Method section. At line 122, the word “too” could read as “tool”. There is need for a specific paragraph describing the objective and research question for this study.

Methods

Well described for the process used. However, the study design is not clear. Is it a qualitative, quantitative, or mixed? It does not give details on how the data were analysed. Any test used? Statistical software? Database created with what software (STATA, SPSS, … Excel,…)? Was there a needed sample size for this study? How was this calculated?

By reading the following paragraph, we can conclude that it was a mixed method: A thematic analysis (i.e., inductive, and semantic approaches) was used to group, analyze, and describe major findings. Further descriptive statistics (percentages, tables, charts, etc.) were used to summarize the quantitative data.

However, when reading the table 1, it shows figures. The design should be clearly stated and described in detail. The thematic approach used should be clear as the manuscript is already written, it should not say i.e.

The Ethics statement is not detailed. As this includes live participants who responded to questions that were included in the analysis, there is a need to know if these participants gave their consent for their answers to be involved in any publication. Were they aware that these questions were asked for the purpose of research?

Results

The lines 153 to 155 should be in the “Methods” section. The lines 170 to 189 should be in the “Methods” section. The following sentence: In addition to this Africa, CDC had IT support… should read as: In addition to this, Africa CDC had IT support… At the line 204, it is good to clarify what the additional question is. In the table 1 (line 206), why percentages of No answers are in italic while they are not for Yes?

Line 210: The following sentence: In the individual webinar sessions, there were, on average, 26 interactions between the participants and facilitators, ranging from 4 to 56 interactions per session.

should read as: On average, there were 26 interactions between the participants and facilitators in the individual webinar sessions, ranging from 4 to 56 interactions per session.

It could be better to list some examples of best practices that were exchanged between countries through the webinars and understand how these were implemented where they are lacking to improve the response to the pandemic. Please include some practices that each PHEOC learned from each other.

Line 273: The sentence: Organizers recognized that participants tended to drop off the webinar after one hour, likely because of conflicting commitments, therefore, it is recommended that future webinar sessions do not extend beyond one hour.

should be split into 2 and should read as Organizers recognized that participants tended to drop off the webinar after one hour, likely because of conflicting commitments. Therefore, it is recommended that future webinar sessions do not extend beyond one hour.

Line 293: This sentence is too long: The success of the topic choice and strong communication of webinar sessions in advance is indicative of the commitment of the PHEOC Network partners, PHEOC Webinar working group, and secretariat, and emphasizes the effort involved in conducting a weekly, interactive webinar series, especially when hosted by SME’s whom themselves were engaged in COVID-19 response activities.

and can split into 2 as: The success of the topic choice and strong communication of webinar sessions in advance are indicative of the commitment of the PHEOC Network partners, PHEOC Webinar working group, and secretariat. They emphasize the effort involved in conducting a weekly, interactive webinar series, especially when hosted by SME’s whom themselves were engaged in COVID-19 response activities.

Line 304: This sentence is too long: The interactions observed appear to be low as compared to the number of attendees per session (235 on average), which could be due to the limited time allocated for discussion, however other considerations may include the likelihood of attendees listening to the session while conducting other work, language barriers or their ability or confidence to engage with the other attendees and facilitators in a virtual environment.

and can split into 2 as: The interactions observed appear to be low as compared to the number of attendees per session (235 on average), which could be due to the limited time allocated for discussion. However, other considerations may include the likelihood of attendees listening to the session while conducting other work, language barriers or their ability, or confidence to engage with the other attendees and facilitators in a virtual environment.

Conclusion

Line 321: This sentence is too long: Besides, there was an overall low attendance rate though the high number of registrants indicates interest and acceptance of the Webinar Series, and almost half of the participants came from just a few countries, possibly explained by biased in the advertisement of the webinars based on partners involved.

and can split into 2 as: Besides, there was an overall low attendance rate though the high number of registrants indicate interest and acceptance of the Webinar Series. Almost half of the participants came from just a few countries, possibly explained by biased in the advertisement of the webinars based on partners involved.

It could good to see how this program can be made sustainable and improved to serve as model for future health emergencies.

Reviewer #4: I am using the "Revised Version" of the manuscript for this review and am editing the text based on the accompanying Line Numbers.

GENERAL COMMENTS:

This article describes an innovative set of interventions to improve staff knowledge and overall public health capacity in Africa, as a response to the novel COVID-19 pandemic. It provides relevant information on the processes, and the evolving approaches over time, in response to participant feedback.

A major statistical issue that needs to be addressed is the percentage level of participation. The authors need to be more clear and precise as to what is the denominator that is being used to estimate the different types of participation. In the paragraph titled Webinar participation (Lines 228 to 242) you state that a total of 95,230 participants registered the the Webinar interventions. I assume that this is the denominator that measures levels of participation for many of the citations. However, this explanation doesn't appear until late in the paper. In addition, can you better define this? It would be helpful to describe how you advertised this approach broadly and how various public health persons signed onto an active e-mail list that was then used to notify participants of upcoming sessions and the availability of materials. I would recommend that you provide a short summary in the ABSTRACT of how this overarching denominator was established and then at the beginning of the METHODS section provide more detail. The relevant denominator becomes confusing, because in the Abstract and early in the Main article you cite that 112,354 persons interacted with Zoom and other webinar resources. Were these discrete individuals that had enrolled onto an e-mail contact list? Many readers will confuse this number with the registered participants number. I would recommend that you remove the 112,354 citation in the abstract and instead present the number of persons who registered for the approach (95,230)

Another major issue is that the article relies on participant self assessment to address the overall impact of the approach. It would be helpful in the Discussion session to briefly consider and present possible methods in future evaluations of this approach for a more impartial assessment as to whether public health services were actually improved. Did the quantity and quality of services get better and in what ways? Was there any impact on incidence, standards of care, patient outcomes, etc, etc??

SPECIFIC COMMENTS:

I have made multiple recommendations below, using the line numbering, to address spelling, grammatical, and technical issues.

ABSTRACT

In Line 40, replace the word, "through" with "utilizing."

In Line 54-55, the final phrase is awkward and needs to be rewritten. For example, "...and the simplicity of the delivery of the training, through interactive webinairs and a community of practice, encouraged a greater number of public health staff to participate, and to spread the word of the training to their own networks."

In Line 45 remove the sentence, " Major findings were synthesized and described per thematic area". You have duplicated the previous sentence.

In Lines 48 to 50 the authors refer to "112,354 persons interacted with the sessions across ZOOM, Facebook and You tube......" This is confusing because the reader may assume that this is a denominator for participation. But it isn't. The denominator is actually 95,230 participants that were registered to be part of the approach. Per my note at the beginning of this Comments section, it would be helpful if you provide information on the denominator for participation earlier in the document and remove the 112, 354 number from the Abstract section.

INTRODUCTION:

In lines 66-67, revise to "...abruptly interrupted, testing the recent increases in health security investments that African countries had received since the end of the West African Ebola outbreak in 2016."

Line 92, add "events" after face-to-face"

Line 93, replace "global" with "globally"

Line 94 replace "till" with "still."

In line 97, replace "suing" with "using."

In line 99, replace "...consiting of creating and online..." with "and included the creation of a Community of Practice (CoP, comprising mainly PHEOC professionals fro the African continent."

In Line 104, can you footnote the term "Discord" and provide a short description of the application?

METHODS

In Lines 120 to 123, can you add any information on the approximated number of staff in the secretariat and any information on the annual budgeting costs?

WEBINAR FACILITATION.

Line 125, rewrite, "The webinar series was initiated to respond to identified gaps in multiple....."

Line 127, can you find a different term for "take -off curriculum?" I don't know what this means.

Line 137, replace "tour" with "tours" and make "human resource" plural-resources"

Line 145, replace "series" with "multi-part series"

Liness 154-155. Use this sentence which describes DISCORD, as a footnote as noted above for Line 104.

Line 198, replace the word topic with the plural, "topics"

Lines 222-226, as stated in my general comments, the authors do not define the baseline numbers, so the participation percentages are not clear. Please provide the definition of and the actual number of persons who make up the audience for this exercise. Is there an enrollee email address list that was created??? How was this done?

Line 244, add the word, "with" just before the words, "...webinar 16,..."

DISCUSSION

Overall, I found the DISCUSSION section to be excellent. However, there are some key sections that could be improved.

Lines 307 to 310. This sentence is very confusing. Could you replace it with the following, "However, , the inclusion of key partners of the Africa CDC and US CDC in the PHEOC Webinar Series working group is likely to have resulted in an inordinately large number of of participants from outside the Africa PHEOC network."

Line 320, what is an "MS PHEOC?"

LIMITATIONS

Lines 367 to 370. I found this section to be extremely confusing. The authors are trying to describe possible biases in the topic selection, but these arguments are not well presented. Can you rewrite this section??

Lined 384. In this sentence the authors refer to the PHEM training. What does the abbreviation PHEM mean?

Again, overall I think the paper is good and deserves to be published. However, if the above revisions are incorporated the paper will be more accessible to the readers.

Reviewer #5: Great paper. I have recommended acceptance by the journal - however, please do take the time to integrate the suggestions I have made the the attached document.

7. PLOS authors have the option to publish the peer review history of their article (what does this mean?). If published, this will include your full peer review and any attached files.

**Do you want your identity to be public for this peer review?** For information about this choice, including consent withdrawal, please see our Privacy Policy.

Reviewer #3: No

Reviewer #4: **Yes: **Paul R De Lay, MD, DTM&H (Lond)

Reviewer #5: **Yes: **Beth A Tippett Barr

---

## [Decision Letter · Decision Letter 2]

11 May 2023

Strengthening COVID-19 pandemic response coordination through public health emergency operations centers (PHEOCs) in Africa: Review of a multi-faceted knowledge management and sharing approach, 2020-2021

PGPH-D-22-01832R2

Dear Womi Eteng:

We are pleased to inform you that your manuscript 'Strengthening COVID-19 pandemic response coordination through public health emergency operations centers (PHEOCs) in Africa: Review of a multi-faceted knowledge management and sharing approach, 2020-2021' has been provisionally accepted for publication in PLOS Global Public Health.

Before your manuscript can be formally accepted you will need to complete some formatting changes, which you will receive in a follow up email. Please do consider the changes reviewers 4 and 6 describe. A member of our team will be in touch with a set of requests.

Best regards,

Megan Coffee, MD, PhD

Academic Editor

Reviewer Comments (if any, and for reference):

Reviewer's Responses to Questions

**Comments to the Author**

1. If the authors have adequately addressed your comments raised in a previous round of review and you feel that this manuscript is now acceptable for publication, you may indicate that here to bypass the “Comments to the Author” section, enter your conflict of interest statement in the “Confidential to Editor” section, and submit your "Accept" recommendation.

Reviewer #4: All comments have been addressed

Reviewer #6: All comments have been addressed

Reviewer #7: All comments have been addressed

2. Does this manuscript meet PLOS Global Public Health’s publication criteria? Is the manuscript technically sound, and do the data support the conclusions? The manuscript must describe methodologically and ethically rigorous research with conclusions that are appropriately drawn based on the data presented.

Reviewer #4: Yes

Reviewer #6: Yes

Reviewer #7: Yes

3. Has the statistical analysis been performed appropriately and rigorously?

Reviewer #4: Yes

Reviewer #6: N/A

Reviewer #7: Yes

4. Have the authors made all data underlying the findings in their manuscript fully available (please refer to the Data Availability Statement at the start of the manuscript PDF file)?

Reviewer #4: Yes

Reviewer #6: Yes

Reviewer #7: Yes

5. Is the manuscript presented in an intelligible fashion and written in standard English?

Reviewer #4: Yes

Reviewer #6: Yes

Reviewer #7: Yes

6. Review Comments to the Author

Reviewer #4: This article presents an innovative set of interventions to improve staff knowledge and public health capacity, as a response the novel COVID-19 pandemic. All of my previous recommended revisions and suggestions for additional material have been accepted and have been adequately addressed. I recommend that the article be published. There were a few minor errors that should be corrected:

Line 76, the word "...standard..." should be "...standards...."

Line 275, the word"...documentations.." should be "documentation..."

Line 309, replace the words, "...carter for..." with "..respond to..."

Reviewer #6: Review Comments: Abstract

1. The title should be modified to ‘Impact of Knowledge Management and information sharing during COVID-19 pandemic response coordination using Public Health Emergency Operation Centre [PHEOC] as a pivot’

2. Abstract should not exceed 250 words

3. Restructure the aim of the study to reflect the impact or clarify if it is about building capacity of the attendees

4. Your methods should be clearly stated in a concise manner. What type of study is this? although how it was conducted was actually stated

5. Your conclusion should be based on the findings documented, was there any evidence may be after 3 months of stopping the webinar if the capacity has been grown or built as the case may be or was there any suggestions as to improvement in coordination mechanism at the PHEOC?

6. What does your recommendation aim to achieve?

Reviewer #7: (No Response)

7. PLOS authors have the option to publish the peer review history of their article (what does this mean?). If published, this will include your full peer review and any attached files.

**Do you want your identity to be public for this peer review?** For information about this choice, including consent withdrawal, please see our Privacy Policy.

Reviewer #4: **Yes: **Paul R De Lay, MD, DTM&H (Lond)

Reviewer #6: No

Reviewer #7: No
